Diversity and structure of the microbial community in rhizosphere soil of Fritillaria ussuriensis at different health levels

Jiao Ning 1
Song Xiaoshuang 2
Song Ruiqing 1
Yin Dachuan 3
Deng Xun dengxun@nefu.edu.cn 2
1 College of Forestry, Northeast Forestry University , Harbin , China
2 Institute of Forestry Protection, Heilongjiang Forestry Academy , Harbin , China
3 College of Forestry, Shenyang Agricultural University , Shenyang , China
Kang Xiaoming
Electronic publication date: 2022 Jan 25
Publication date: 2022
Volume: 10
Electronic Location ID: e12778
Received 2021 Aug 17; Accepted 2021 Dec 21
Copyright: ©2022 Jiao et al.
Copyright year: 2022
Copyright holder: Jiao et al.
License: This is an open access article distributed under the terms of the Creative Commons Attribution License, which permits unrestricted use, distribution, reproduction and adaptation in any medium and for any purpose provided that it is properly attributed. For attribution, the original author(s), title, publication source (PeerJ) and either DOI or URL of the article must be cited.
License URL: https://creativecommons.org/licenses/by/4.0/

Keywords: Fritillaria wilt disease, Soil microbial community, Illumina MiSeq, Health level

Funding: National Key Research and Development Program 2016YFC0500308-08 National Natural Science Foundation of China 31670649 31700564 31170597 31200484 This research was funded by (National Key Research and Development Program) grant number (2016YFC0500308-08), (National Natural Science Foundation of China) grant number (31670649, 31700564, 31170597,31200484). The funders had no role in study design, data collection and analysis, decision to publish, or preparation of the manuscript.

==============================
Fritillaria wilt is a kind of soil-borne disease that causes a large reduction in the yield of Fritillaria ussuriensis. The diversity and structure of the soil microbial community are important factors affecting the health of Fritillaria ussuriensis. The analysis of the microbial community in the diseased and healthy soils provided a theoretical basis for revealing the pathological mechanism and prevention of Fritillaria wilt disease. In the present study, we sequenced the soil microorganisms from healthy (H), pathology (P) and blank (B) soil samples by Illumina MiSeq. Determined the soil physicochemical properties respectively, analyzed the soil microbial diversity and structure, and constructed single factor co-correlation networks among microbial genera. The results showed that Ascomycota (48.36%), Mortierellomycota (23.06%), Basidiomycota (19.00%), Proteobacteria (31.74%), and Acidobacteria (20.95%) were dominant in the soil. The diversity of healthy soil was significantly greater than that of diseased soil samples (P and B) (P < 0.05). The populations of Fusarium and Humicola significantly increased in the diseased soil sample (P and B) (P < 0.05). RB41 (4.74%) and Arthrobacter (3.30%) were the most abundant genera in the healthy soil. Total nitrogen (TN), available nitrogen (AN), total potassium (TK), available potassium (AK), and inorganic salt (salt) were significantly correlated with soil microbial communities (P < 0.05). The relationship between fungi and the plant was mostly positive, whereas bacteria showed the opposite trend. In conclusion, the diversity and structure of the soil microbial community were closely related to the health level of Fritillaria ussuriensis. Fusarium and Humicola affect the severity of Fritillaria wilt disease, while RB41 and Arthrobacter are the important indicators for maintaining the health of Fritillaria ussuriensis. Moreover, environmental factors greatly affect the abundance and formation of soil microbial community. The interactions in microbial communities also influence the healthy growth of Fritillaria ussuriensis.

Introduction

Fritillaria ussuriensis Maxim., also known as Fritillaria ussuriensis, is a perennial herb belonging to the genus Fritillaria of the family Liliaceae. The dried bulb of Fritillaria ussuriensis has heat-clearing, detoxicating, cough-relieving, and phlegm-resolving effects. It is an important medicinal material in Northeast China (Park et al., 2017). Fritillaria ussuriensis is mainly distributed Changbai Mountains and the southern part of Xiaoxing’an Mountains in China (Day et al., 2014). The wild Fritillaria ussuriensis has been seriously damaged, therefore, which has been artificially cultivated on a large scale in Northeast China (Xu et al., 2013). Relevant studies have shown that Fritillaria ussuriensis should plant in black soil with sunny leeward, flat terrain, sufficient water and good drainage (Ding et al., 2018). However, Fusarium is the main cause of Fritillaria wilt, a soil-borne disease, of Fritillaria ussuriensis in the long-term continuous cropping. It spreads in the soil and even causes the failure of crop production in serious cases (Baayen et al., 2001). In order to study the diversity and structure of microbial communities in the rhizosphere soil of Fritillaria ussuriensis at different health levels, we selected Fritillaria ussuriensis Planting Site of Hailin Forestry Farm which have suitable planting conditions.

It is of great importance to study the rhizosphere soils of diseased and healthy plants in long-term continuous cropping to maintain the ecological balance and provide the economic benefits for sustainable development. As far as we know, rational use or improvement of fields based on soil characteristics, soil microbial diversity and structure can manage crops more efficiently. Soil microbial community plays a key role in managing soil fertility, nutrient cycles, and plant health (Fierer et al., 2012), which are directly related to herbaceous plant health. In the soil ecosystem, the environment in close proximity to plant roots is rhizosphere, a dynamic habitat supporting resource exchange between plants and the soil environment (Peiffer et al., 2013). The microbial community in the rhizosphere soil is considered as the second genome of plants, which changes under the influence of plant roots (Philippot et al., 2013). At the same time, the diversity and colonization ability of soil microbial communities in different microhabitats affect the growth rate of pathogens and also play an important role in improving plant health (Li et al., 2013; Kloepper & Beauchamp, 1992).

Soil conditions can indirectly reflect the level of plant health, and the occurrence of plant soil-borne diseases is considered to be an unstable and unhealthy state of soil micro-ecology (Doran, Sarrantonio & Liebig, 1996; Karlen et al., 1997). Studies have shown that changes in soil microbial community diversity and structure will affect the occurrence of soil-borne diseases, soil microbial communities are very sensitive to diseased soils, and there are very few pathogens in soils with high microbial diversity (Shiomi et al., 1999; Benizri et al., 2005; Pérez-Piqueres et al., 2006). Bulluck III et al. (2002) improved soil microbial diversity by applying organic fertilizers, the reproduction density of plant pathogens Phytophthora and Pythium has been greatly reduced. Reports have pointed out that the soil microbial community structure can be used to assess soil disease resistance and plant health, and some characteristic indicator microorganisms that inhibit disease can be found from soil microbial community (Gong et al., 2007). On the other hand, the soil-borne diseases usually reduce soil microbial diversity, Yang, Crowley & Menge (2001) compared the bacterial community structure in the avocados rhizosphere soil infected by Phytophthora cinnamani and found that the bacterial community structure of healthy soil was similar, while the soil infected by Phytophthora cinnamani was significantly different, and the bacterial diversity index of the infested soil also decreased significantly. Previously, in the field evaluation of root rot disease in Fritillaria ussuriensis, we found that the abundance and diversity of soil microbial community decreased, while the population of pathogens in the healthy soil sample was quite low, which would not cause harm, indicating that the microbial community structure affected the health of Fritillaria ussuriensis (Song et al., 2016). In this study, we systematically analyzed the diversity and structure of soil microbial communities in the rhizosphere of Fritillaria ussuriensis at different health levels to illustrate the impact of changes in the microbial community on the Fritillaria ussuri ensis health level.

The soil-borne disease is also closely related to environmental factors (Daguerre et al., 2014). Most studies have shown that soil pH is negatively correlated with soil disease resistance (Rimé et al., 2003), soil viscosity can help improve plant disease resistance (Duffy, Ownley & Weller, 1997), the form of soil nitrogen also affects soil disease resistance and ammonia nitrogen is beneficial to soil disease resistance (Tenuta & Lazarovits, 2004), soil available potassium content is also negatively correlated with soil disease resistance (Xu et al., 2004). Xu, Sun & Wang (2009) showed that the ginger skin rot disease was more serious in soil with high organic matter quality. Certainly, the soil physicochemical properties also directly affect the diversity and structure of soil microbial community (Doi & Ranamukhaarachchi, 2009). Current researches have shown that soil organic matter has been proven to be a key factor affecting the diversity and structure of soil microbial communities (Sessitsch et al., 2001). For example, Yan, McBratney & Copeland (2000) found that soil microbial diversity was significantly related to soil pH and organic matter. Zhao, Fang & Tian (2007) studied the relationship between environmental factors and soil microbes in Chinese fir plantations and found that the total number of microbes was significantly positively correlated with soil organic carbon content and total nitrogen content. There have been a lot of researches on the direct influence of environmental factors on microorganisms, but there is little research on the correlation between environmental factors and soil microorganisms in the plant pathological process. What are the main factors affecting the microbial diversity in rhizosphere soil of Fritillaria ussuriensis at different health levels, and how their interaction mechanism has not been reported. The solution of these problems will be of great significance to the use of microbial diversity to regulate the soil micro-ecosystem and improve the disease resistance of Fritillaria ussuriensis.

Material and Methods

Site depiction and sampling

The experiment location is at the Planting Site, in which Fritillaria ussuriensis is cultivated (N 44°89′−44°88′, E 129°30′−129°31′), in Hailin Forestry Farm, Mudanjiang City, PRC. At the Changbai Mountains, this site is the key planting base for crops in Northeast China. The area has monsoon and middle-latitude climates, with a yearly mean temperature of 2.2 °C and a yearly mean rainfall of 550 mm. The persistent sequential cropping was employed in the plantation, in which the average altitude is 550 m, the soil is meadow dark brown soil, the average soil water content is 24.55%, the average temperature of the 0∼5 cm soil layer is 21.6 °C, the average soil density is 1.54 g/cm3, and the average soil porosity is 42.01%. There is 30 years history of artificial cultivation in Planting Site of Fritillaria ussuriensis, the total cultivation area is nearly 2 km2. The same agronomic management practices and fertilization system were adopted at the experimental site, but years of continuous cropping resulted in serious soil-borne diseases including wilt disease. Through the preliminary investigation, we found that the wilt diseases severity of Fritillaria ussuriensis in different cultivation plots was different (Fig. 1). The growth cycle on aboveground parts of the Fritillaria ussuriensis is from April to June, therefore, the time of our investigation and sampling was in late May 2018 (late spring). When Fritillaria ussuriensis wilt disease occurs, bulbs rotted and turned black underground, and the aboveground plants appeared withered, obvious patches were formed on the soil surface of Fritillaria ussuriensis (pathology soil). The wilt disease continued to develop, and obvious empty window plots were formed in the second year (blank soil), while healthy plots grow vigorously, and there are no wilting patches and blank plots (healthy soil). Finally we distinguished and collected rhizosphere soil samples (healthy, pathology and blank soil) with different health levels according to the pathological condition of Fritillaria ussuriensis wilt disease at the the Planting Site in Hailin Forestry Farm (Fig. 2). The sampling method was as follows: we used the shovel to dig the soil profile at 15 cm depth, collecting the rhizosphere soil that was not removed after shaking, and then transferred it into sterile sample bags for the experiment. 10 sampling points in planting plots with different health levels were randomly selected respectively, after mixing the soil samples with the same health level, 6 repeated treatments were collected for each of the 3 different healthy levels soil samples (healthy, pathology and blank soil) respectively, of which 3 repeated treatments were used for soil fungal sequencing and the other 3 were used for soil bacterial sequencing, for a total of 18 (3 × 6 = 18) soil samples. All samples were homogenized thoroughly through a 2 mm sieve, transferred to the lab on ice, and afterwards reserved at −80 °C for Illumina MiSeq. Meanwhile, the samples used to determine the soil physicochemical properties were stored in a dry place after air-drying (Collignon et al., 2011; Shanmugam et al., 2011; Uroz et al., 2016).

Figure 1 Locations of the rhizosphere soils from Fritillaria ussuriensis in different health levels.

Sampling point was located in Hailin Forestry Farm, Hailin City, Mudanjiang City, Heilongjiang Province, China. The healthy samples were located at 44.8942 north latitude and 129.3139 east longitude; the pathology sample was located at 44.8917 north latitude and 129.3173 east longitude; the blank sample was located at 44.8911 north latitude and 129.3167 east longitude.

Figure 2 Planting survey plot of Fritillaria ussuriensis in different health levels.

Determination of soil physicochemical properties

Soil pH was determined in a soil-to-water (1:2.5, W/V) mixtures of dry soil and distilled water using a HACH HQ30d pH meter (BANTE, Shanghai, China). Soil organic matter (OM) was determined by the potassium dichromate heating method. Total nitrogen (TN) content was measured with Kjeldahl digestion and distillation azotometry (Kjeldahl distillation unit K9840, Shandong, China). Available nitrogen (AN) content was measured by MgO steam distillation method. Total phosphorus (TP) content was measured by the Mo-Sb colorimetric method. Available phosphorus (AP) content was measured by lixiviating-molybdenum blue colorimetry after extraction with 0.5 M NaHCO3 (pH=8.5) for 30 min. Total potassium (TK) content was measured by sodium hydroxide fusion-flame spectrophotometer method and available potassium (AK) content was measured by NH4OAc extraction-flame spectrophotometer method (flame photometer FP6410, Shanghai, China). Total soluble salt (salt) was determined by residue weighing method (set dry soil to distilled water = 1:5, W/V) (Bao, 2000).

Soil DNA abstraction, PCR enlargement, and Illumina MiSeq sequence

The entire soil DNA was abstracted via the E.Z.N.A® soil DNA Kit (Omega BioTek, America). NanoDrop ND-2000C (Thermo, America) was employed to identify the DNA level and purity, while 1% gel electrophoretic method was adopted to assess the quality of DNA (Chen et al., 2018; Zhou et al., 2016). The primer sets including ITS1F (5′-CTTGGTCATTTAGAGGAAGTAA-3′) and ITS2R (5′-GCTGCGTTCTTCATCGATGC-3′) were selected to target the ITS1-ITS2 region for the characterization of fungal communities. 338F (5′-ACTCCTACGGGAGGCAGCAG-3′) and 806R (5′-GGACTACHVGGGTW TCTAAT-3′) primers were adopted for the amplification of the V3-V4 hyper variant areas for the bacterium 16S rRNA gene (Mori et al., 2013; Xu et al., 2016). The reactive activities were completed in triplicate via a 20 µL reactive mixed solution, involving 4 µL 5 ×FastPfu Buffer, 2 µL 2.5 mM dNTPs, 0.8 µL every primer (5 µM), 0.4 µL FastPfu Polyase, 10 ng template DNA, 0.2 µL of BSA, and 11.6 µL of redistilled water (Wang,Chen & Zhang, 2017; Fu, Zhang & Hou, 2019). The prerequisites of amplified PCR: posterior to the incipient denaturating at 95 °C for 3 min, PCR was completed for 27 cycles at 95 °C for 30 s, annealed at 55 °C for 30 s, elongated at 72 °C for 45 s, and an eventual elongation at 72 °C for 10 min (PCR: ABI GeneAmp® 9700, USA). The PCR results were treated with purification via a PCR Purifying Kit (Axygen Bio, America). QuantiFluor™ -ST (Promega, America) was employed for quantitation determination. Based on the requirements of sequencing, specimens were gathered proportionally, and a FastPfu database was built. Eventually, the sequencing of the purification libraries were realized via the Illumina MiSeq (TruSeqTM DNA Specimen Preparation Kit, America) (Yang et al., 2017a; Yang et al., 2017b).

Bioinformatics analysis

Posterior to the removing of the adapters and primer sequences, original sequences were merged as per the distinctive stripe code via QIIME (Caporaso et al., 2010). The split sequences for every specimen were assembled via FLASH 1.2.7 (Magoc & Salzberg, 2011), and short sequences (Seq < 200 bp) and low-quality sequences (Q < 0.5) were filtered out. The UCHIME approach was adopted for the removal of chimera sequences (Edgar et al., 2011). The UPARSE 7.1 arithmetic with a 97% sequential similarity threshold (St) was employed to acquire OTUs (Edgar, 2013). The UNITE database 7.0 was used to annotate fungal OTUs (Kõljalg et al., 2013), and the RDP database (version 9) was employed to classify bacterial OTUs (Wang et al., 2007), with the confident liminal value of 80%. The alpha variety assay was completed via Mothur 1.30 (Buée et al., 2009). Based on the Bary-Curtis algorithm, the beta diversity was analyzed using PCoA (Lozupone, Hamady & Knight, 2006). The diversity in the microbiological population was contrasted via AMOVA (Analysis of molecular variance) (Meirmans, 2006).

Network analysis

Divided healthy, pathology and blank soil samples into 3 groups, and comprehensively analyzed. The top 25 abundant fungal and bacterial OTUs were selected from the soil samples. NetworkX was used to obtain the related information from different genera and construct interaction networks (Klarner, Streck & Siebert, 2016). The Random Matrix Theory (RMT) was employed to automatically identify the appropriate similarity threshold (St) before network construction. At last, the JavaScript software was used to create the single factor co-correlation network.

Statistical analyses

SPSS 22.0 (IBM, America) was used for statistic assay. The one-way ANOVA and the Duncan’s (α = 0.05) test were adopted to contrast the soil physicochemical properties and microbial level and diversity (Halifu et al., 2019). Pearson’s correlative was adopted for the establishment of an association among microbe genera, environment factors, and alpha variety (Zhou et al., 2017).

Sequence registration numbers

The sequence data were deposited in the NCBI Sequence Read Archive (SRA) database with the accession number of SRR13288220 –SRR13288237.

Results

Soil physicochemical properties

The soil physicochemical properties of Fritillaria ussuriensis at different health levels are summarized in Table 1. The average pH in diseased soil (P and B) was significantly lower than that in healthy soil (P < 0.05). There was no significant difference in pH between pathology and blank samples, indicating that the soil pH had a lower correlation with the severity of the disease. The content of organic matter (OM) was the highest in the blank sample (P < 0.05). Moreover, the contents of total nitrogen (TN), available nitrogen (AN), total phosphorus (TP), and available phosphorus (AP) increased significantly and then decreased slightly with increasing the severity of disease (H →P →B), while the contents of total potassium (TK), available potassium (AK), and total soluble salt (salt) increased significantly (P < 0.05).

Table 1 Comparative analysis of soil physicochemical properties of Healthy, Pathology and Blank.

Sample	pH	OM
g/kg	TN
g/kg	AN
mg/kg	TP
g/kg	AP
mg/kg	TK
g/kg	AK
mg/kg	salt
g/kg	
Healthy	7.33 ± 0.32A	22.33 ± 1.03B	5.64 ± 0.88C	18.13 ± 0.47B	2.04 ± 0.16B	76.25 ± 1.42B	7.03 ± 0.29B	204.12 ± 11.42C	0.83 ± 0.03C	
Pathology	6.24 ± 0.95B	21.26 ± 0.99B	7.31 ± 0.05A	21.26 ± 1.00A	3.01 ± 0.42A	95.52 ± 5.14A	7.38 ± 0.32B	230.49 ± 18.03B	1.08 ± 0.06B	
Blank	6.31 ± 0.31B	26.17 ± 0.70A	6.46 ± 0.48B	20.47 ± 0.66A	2.11 ± 0.55B	80.39 ± 0.70B	8.70 ± 1.10A	310.22 ± 5.00A	1.41 ± 0.12A	
Notes.

Different letters indicate a significant difference at P < 0.05 according to Duncans new multiple range test.

OM organic matter

TN total nitrogen

AN available nitrogen

TP total phosphorus

AP available phosphorus

TK total potassium

AK available potassium

salt total soluble salt

Diversity of microbial community

A total of 451,644 and 399,003 valid fungal and bacterial sequences, respectively, were obtained from 18 healthy, pathology, and blank soil samples, with the average sequence lengths of 240.83 bp for fungi and 417.52 bp for bacteria. The microbial sequences of soil samples were clustered into 1,810 fungal and 3,737 bacterial OTUs at the 97% identity threshold after splitting and removing redundancy. Valid sequences were randomly sampled, and the OTUs from the extracted sequences were used to construct the rarefaction curves (Fig. S1). The rarefaction curves of fungal and bacterial OTUs changed smoothly, which indicates that the information about microbes from the samples was fully obtained. This confirms the validity of the study on soil fungal and bacterial communities.

The Venn diagrams showed the OTU level for the microbial community of soil samples (Fig. 3). The number of fungal OTUs in healthy samples was the highest, while blank samples had the lowest number. But the distribution of bacterial OTUs was the opposite of fungal OTUs. The number of bacterial OTUs was the highest in blank samples, while it was the lowest in healthy samples. The highest number of OTUs was found in healthy samples, for both fungal and bacterial communities, with 53 and 19 OTUs, respectively. The number of shared OTUs in all samples was the highest, with 50.77% of fungal OTUs and 80.06% of bacterial OTUs. The above-mentioned distribution of OTUs showed that the fungal OTUs level in pathology samples was more similar to that in blank samples, and there was a significant difference in OTUs compared with healthy samples, while the bacterial OTUs level was found to be similar in all samples.

Figure 3 Venn diagram showing the shared operational taxonomic units (OTUs) of Healthy, Pathology and Blank.

The column chart shows the size of each list and the Bar chart shows the number of single or multiple elements.

ANOVA for the α diversity of soil samples at different health levels indicated the coverage for soil fungal samples with coverage over 99% and close to 98% for soil bacterial samples, which coincided with rarefaction curves of OTUs (Table 2). The Shannon index of fungal and bacterial communities decreased significantly and then increased with increasing the severity of disease (H →P →B) (P < 0.05). The Simpson index of fungal communities in healthy samples was significantly higher than that in diseased soil (P and B), while it hardly varied for the bacterial community during the pathological process of Fritillaria ussuriensis (P < 0.05). In the stage of healthy to pathology, the Ace index and Chao index of fungal and bacterial communities decreased slightly, while the difference between pathology and blank was not significant. These findings showed that the diversity of fungal and bacterial communities in pathology samples was lower than in healthy or blank samples. However, there was no significant difference in diversity between healthy and blank samples.

Table 2 Soil α diversity index of fungal and bacterial communities in Healthy, Pathology and Blank.

	Sample	Shannon	Simpson	Ace	Chao	Coverage	
Fungi	Healthy	4.49 ± 0.09A	0.031 ± 0.00B	796.81 ± 68.43A	802.3 ± 70.75A	99.84 ± 0.00%A	
Pathology	4.07 ± 0.07B	0.051 ± 0.00A	750.59 ± 61.03AB	751.89 ± 61.34AB	99.73 ± 0.00%A	
Blank	4.27 ± 0.14B	0.044 ± 0.00A	806.59 ± 65.60A	815.22 ± 66.59A	99.84 ± 0.00%A	
Bacteria	Healthy	6.59 ± 0.06a	0.0039 ± 0.00a	3057.47 ± 61.69a	3044.13 ± 63.73a	98.13 ± 0.00%a	
Pathology	6.46 ± 0.01b	0.0037 ± 0.00a	2943.83 ± 72.98ab	2895.71 ± 86.24b	97.86 ± 0.00%a	
Blank	6.54 ± 0.03ab	0.0038 ± 0.00a	3029.25 ± 19.63a	2992.48 ± 6.61ab	97.94 ± 0.00%a	
Notes.

Different letters indicate significant difference at P < 0.05 according to Duncan’s new multiple range test.

Soil microbial community structure and composition

To further compare the variations in the structure of fungal and bacterial communities in healthy, pathology, and blank samples, based on the Bary-Curtis algorithm, the principal coordinate analysis (PCoA) was employed (Fig. 4). Diseased samples (P and B) were separated from the healthy samples, indicating large differences in the structure of microbial communities at different health levels. In addition, the degree of dispersion in fungal communities between pathology and blank samples were smaller, on the contrary, the bacterial communities were larger, suggesting that differences in the structure of fungal and bacterial communities between diseased samples (P and B).

Figure 4 Principle coordinate analysis (PCoA) of fungal and bacterial communities structures in Healthy, Pathology and Blank.

To verify the differences observed in fungal and bacterial communities at different health levels, the relative abundances of different phyla and genera from the rhizosphere soil samples were compared (Figs. 5 and 6). Average OTUs of 9 samples of fungus were classified into 13 phyla, 35 classes, 80 orders, 195 families, 325 genera, and 476 species. The phyla of fungi mainly included Ascomycota (48.36%), Mortierellomycota (23.06%), and Basidiomycota (19.00%), accounting for more than 90% of the abundance. The abundance of Ascomycota was the highest among all samples at different health levels and almost similar in healthy and pathology samples, with a significant increase in the blank sample. The abundance of Mortierellomycota increased significantly in diseased samples (P and B), but the difference was not significant between pathology and blank samples. The abundance of Basidiomycota decreased significantly with increasing the severity of disease (H →P →B), with the lowest abundance in the blank sample. Average OTUs of 9 samples of bacteria were classified into 34 phyla and 652 genera, including mainly Proteobacteria (31.74%), Acidobacteria (20.95%), Actinobacteria (14.11%), Chloroflexi (8.52%), Bacteroidetes (6.64%), Gemmatimonadetes (5.48%), Patescibacteria (2.97%), Firmicutes (2.68%), Verrucomicrobia (2.08%), Rokubacteria (1.37%), and Latescibacteria (0.56%). We found higher abundances of Proteobacteria, Gemmatimonadetes, and Firmicutes in diseased samples (P and B). The abundance of Acidobacteria decreased significantly with increasing the severity of disease (H →P →B), while the abundances of Chloroflexi and Verrucomicrobia were the highest in the healthy sample.

Figure 5 Relative abundances of the main fungal and bacterial phyla of the rhizosphere soil in Healthy, Pathology and Blank.

The “Others” and “Unclassified_k_Fungi” comprised the unclassified and low-abundance phyla (RA < 0.1%).

Figure 6 Relative abundances of the top seven fungal genera and top 15 bacterial genera of the rhizosphere soil in Healthy, Pathology and Blank.

Different letters represent significance (P < 0.05) of the genus levels in different health levels according to Duncan’s new multiple range test.

To further study the difference in the composition of the soil microbial community at different health levels, the top 7 fungal and 15 bacterial genera with high abundances were selected from the fungal and bacterial communities in healthy, pathology, and blank samples. The difference in the relative abundance at the genus level indicated that Mortierella, Fusarium, Leucosporidium, Mrakia, Guehomyces, Humicola, and Ilyonectria were members of the fungal genera with higher abundances, among which Mortierella exhibiting the highest abundance (22.86%). Compared with the healthy sample, the abundance of Mortierella increased significantly in diseased samples (P and B). The abundances of Fusarium and Humicola increased significantly with increasing the severity of disease (H →P →B), with the blank sample having the highest abundance (15.49% and 5.60%, respectively). On the contrary, the abundances of Mrakia and Guehomyces decreased significantly, with the highest abundance in the healthy sample and more abundance and variation in the bacterial community compared with the fungal community. The highest abundances of RB41 (4.74%) and Arthrobacter (3.30%) were found in the healthy sample, with increasing the severity of disease (H →P →B), the abundance decreased significantly. The relative abundances of Sphingomonas, Bryobacter, Gemmatimonas, Bacillus, Ellin6067, Pedobacter, Acidothermus, and Acidibacter in diseased samples (P and B) were significantly higher than that in healthy samples. Furthermore, the highest abundances of Bryobacter, Acidibacter, Pseudomonas, Massilia, and Haliangium were found in pathology samples.

The relationships between the diversity of soil microbial communities and soil properties

Pearson’s correlation analysis showed that AN and AP were significantly negatively correlated with the Shannon index in the fungal community but significantly positively correlated with the Simpson index (P < 0.05) (Table 3). In the bacterial community, TP and AP were significantly negatively correlated with Shannon index, Ace index, and Chao index, with pH having a significantly positive correlation with Chao index, while AN was significantly negatively correlated with Chao index (P < 0.05).

Table 3 Correlation analyses between diversity indices and soil properties.

	Diversity	pH	OM	TN	AN	TP	AP	TK	AK	salt	
Fungi	Shannon	0.461	0.124	−0.635	−0.721∗	−0.396	−0.834∗∗	−0.119	−0.17	−0.398	
Simpson	−0.611	0.004	0.627	0.822∗∗	0.444	0.803∗∗	0.238	0.364	0.547	
Ace	0.181	0.409	−0.458	−0.261	−0.115	−0.526	0.042	0.214	0.069	
Chao	0.147	0.419	−0.467	−0.238	−0.127	−0.54	0.101	0.238	0.106	
Bacteria	Shannon	0.509	0.093	−0.588	−0.652	−0.670∗	−0.769∗	0.005	−0.103	−0.305	
Simpson	0.034	0.256	−0.485	−0.261	−0.047	−0.169	−0.221	−0.073	−0.072	
Ace	0.596	0.317	−0.586	−0.627	−0.738∗	−0.736∗	0.12	0.039	−0.14	
Chao	0.672∗	0.297	−0.655	−0.736∗	−0.742∗	−0.787∗	−0.034	−0.03	−0.262	
Notes.

* and ** represent significance (P < 0.05 and P < 0.01) of soil samples according to Pearson’s correlation analysis.

About 30 genera of fungal and bacterial communities from rhizosphere soils were significantly different at different health levels. The correlations between the abundances of these genera and pH, OM, TN, AN, TP, AP, TK, AK, salt, and microbial diversity were explored using Pearson’s correlation analysis (Fig. 7). In fungal genera, Thelebolus was significantly negatively correlated with AN, TN, TP, and AP (P < 0.05). Ilyonectria, Acremonium, Cadophora, Gibberella, Mrakia, Guehomyces, and Tetracladium were negatively correlated with TK, AK, AN, salt, and Simpson index (P < 0.05). Moreover, Leucosporidium was significantly positively correlated with AN, TP, AP, and Simpson index, while there was a significantly negative correlation between pH and Shannon index (P < 0.05). Penicillium, Arthrobotrys, Pseudogymnoascus, Fusarium, Nectria, Remersonia, Humicola, Trichoderma, Chaetomium, Mortierella, and Solicoccozyma were positively correlated with OM (P < 0.05), AK, and salt, among which Nectria, Humicola, and Trichoderma also had significant positive correlations with AK (P < 0.001). Chaetomium was significantly positively correlated with the Simpson index. In bacterial genera, Ellin6067, Sphingomonas, Pseudolabrys, Gemmatimonas, and Bacillus were significantly positively correlated with TK, AK, and salt but significantly negatively correlated with pH (P < 0.05). Moreover, there was a significantly positive correlation between Candidatus_Udaeobacter, RB41, and Arthrobacter and pH, while a negative correlation was observed between these genera and TK, AK, and Salt (P < 0.05). RB41 and Arthrobacter were significantly negatively correlated with AK (P <0.001).

Figure 7 Correlation heatmaps between community composition of soil fungi and bacteria composition and environmental factors.

∗, ∗∗ and∗∗∗ represent significance (P < 0.05, P < 0.01, P < 0.001) of soil samples according to Pearson’s correlation analysis.

Fungal and bacterial community single factor co-correlation networks

The microbial single factor network directly showed the complex co-correlation between the rhizosphere soil microbes. The top 25 dominant fungal and bacterial genera were selected and used to construct the relationship network from the rhizosphere soil fungal and bacterial communities. The network topology showed different sizes of networks of fungal and bacterial communities (220 and 320 nodes, respectively) (Table 4). The average connectivity (the average number of connections between each node and other nodes in the network) of 4.4 for fungal and 6.4 for bacterial genera was observed. The average clustering coefficients, which describe how close the neighbors of a node are, were 0.67 and 0.79 for fungi and bacteria, respectively. This indicated that the bacterial network was denser and more complicated than the fungal network. The average path distance in the fungal network (1.86) was slightly longer than that in the bacterial network (1.57), indicating that fungi might interact with each other for a long time. In the fungal network, the significantly higher abundances of Massilia and Fusarium with stronger correlations with other genera were found, indicating that these two genera occupy important niche in the rhizosphere soil of Fritillaria ussuriensis (Fig. 8). In the bacterial network, RB41, Bacillus, Arthrobacter, and Bryoacter all showed a higher correlation, indicating the close interaction between soil bacteria. Compared with the bacterial network, the correlation of fungal network was mostly positive, implying that most pathogenic fungi might cause disease in Fritillaria ussuriensis through a synergistic effect, while it was possible that more bacteria were restricted by each other.

Table 4 Major topological properties of the single factor co-correlation networks of fungal and bacterial communities.

The number of original OUTs was used for network construction by random matrix theory (RMT)-based approach. Network size was the number of nodes in the network. Network diameter was the maximum distance between any two nodes in the network. Transitivity was the probability that two connected nodes of the same node were still connected to each other.

	Number of original OUTs	Network size	Number of genus	Network diameter	Transitivity	Average connect	Average clustering coefficient	Average path distance	
Fungi	400	220	25	4	0.70	4.4	0.67	1.86	
Bacteria	400	320	25	4	0.82	6.4	0.79	1.57	

Discussion

In this study, we compared the physicochemical properties of fungal and bacterial communities of the rhizosphere soil, such as the abundance, diversity, structure, composition, and interactions at different health levels of Fritillaria ussuriensis. The results showed that the resident soil microbial community plays a role in maintaining the health of Fritillaria ussuriensis. The environmental factors were related to the dynamic variation of the microbial community structure. The difference in soil microbial flora and nutrients was an important reason for the occurrence of Fritillaria wilt disease.

Many studies have shown that the microbial diversity of the rhizosphere soil was positively correlated with plant health Yang et al. (2017a). Our study also found that the diversity of the microbial community in the healthy soil sample was higher than that in diseased soil samples (P and B); this research is consistent with the result of Wang,Chen & Zhang (2017) and Yang et al. (2017b) on the microbial community diversity of tobacco wilt disease. It might because the root exudates of healthy plants could provide more nutrients for soil microorganisms, thereby increasing the species richness and diversity of the microbial community (Xuan et al., 2011). Furthermore, we found a negative relationship between the soil microbial diversity and Fritillaria wilt disease (P and B), which can support the idea that microbial diversity is a key factor controlling the pathogen invasion (van Elsas et al., 2012). In addition, beneficial rhizosphere soil microbes occupy space and trophic niches by competing with other counterparts and improve nutrient uptake and plant health and growth by establishing interactions with plant roots (Richardson, 2001). Related studies have shown that the specific changes in soil microbial community diversity and structure were related to differences in soil structure and plant types (Yin et al., 2014; Ding et al., 2019). For example, Chen et al. (2013) found that the microbial diversity of banana wilt disease soil was higher than that of healthy soil. It was because the continuous disease caused changes in the soil physicochemical properties and soil microbial community structures (Xie et al., 2004). In addition, the invasion of pathogenic microorganisms destroyed the original microbial ecological balance in rhizosphere soil, leading to an abnormal increase or decrease of certain microorganisms, the microbial diversity would temporarily increase before the new balance was established (Chi, 1999). Previous studies have shown that the greater growth of the above-ground part of herbaceous plants could provide a large amount of litter for soil microorganisms, and the root system was well-developed, dense in the surface layer of the soil, the root exudates and dead roots were rich energy materials for microorganisms (Smith & Paul, 1990). In this study, the mass deaths of Fritillaria ussuriensis in blank soil produced a large amount of spoilage, which also explained the slightly increased soil organic matter content and soil microbial diversity of Fritillaria ussuriensis in blank soil samples.

Figure 8 Single factor co-correlation networks of fungal and bacterial communities.

The color of the dot in network diagram indicated the phylum category, the size of the dot indicated the abundance of genera, the red line indicated the positive correlation, the blue line indicated the negative correlation, and the thickness of the line indicated the degree of genera correlation.

The PCoA analysis revealed significant variations in the microbial community structure of Fritillaria ussuriensis from the rhizosphere soil at different health levels. We found significant differences in the fungal and bacterial community structure of diseased and healthy soil samples (H, P, and B); this is consistent with the results of previous research conducted by Song et al. (2016) on root rot disease of Fritillaria ussuriensis. The differences in the microbial community structure can be due to different plant root systems, which are consistent with the findings of many previous studies on a key role that plants play in shaping the microbial community structures in the rhizosphere of plants (Philippot et al., 2013; Edwards et al., 2015). Another reason for significant variations in the microbial community structure in soils at different health levels may be significant differences in environmental factors, as soil physicochemical properties have significant impacts on the microbial community structure (Lauber et al., 2008). For example, Wang et al. (2021) found that soil physicochemical properties were important influencing factors driving changes in the number of soil microorganisms of Gastrodia elata. Li, Chen & Wang (2021) compared the differences of the physicochemical properties and microorganisms from low disease soils and high disease soils of strawberry, which found that the total nitrogen content and the spore germination rate of Fusarium oxysporum were extremely significantly negatively correlated. The above showed that environmental factors were the key factors that produce differences in the microbial community structures.

Microbial taxonomic composition strongly varied in rhizosphere soils at different health levels. Ascomycota, Mortierellomycota, and Basidiomycota are the most abundant fungal phyla. Ascomycota and Basidiomycota with high relative abundances are also two common fungal phyla in soils under continuous cropping with vanilla and peanut (Wu et al., 2017; Li et al., 2014). Related research showed that many species of Mortierellomycota can cause plant diseases, and some species of Mortierella can be isolated from stored rotten fruits (Chen, 1992). We found a significant increase in the abundance of Mortierellomycota in diseased samples (P and B), indicating that this fungal phylum might promote the occurrence of Fritillaria wilt disease. In bacterial phyla, except for Proteobacteria with the highest abundance (31.89%), Acidobacteria, Actinobacteria, Chloroflexi, and Firmicutes were relatively abundant, among which Actinobacteria and Firmicutes are known to produce high levels of secondary metabolites and participate in the decomposition, transformation process, and carbon deposition in the rhizosphere (Palaniyandi et al., 2013; Kim et al., 2011). Previous studies found that higher abundances of Actinobacteria and Firmicutes can cause the effective inhibition of Rhizoctonia (Mendes et al., 2011), while Acidobacteria, Chloroflexi, and Bacteroidetes are mainly involved in the decomposition of organic matter (Ai et al., 2015). The results indicated that the variations in the abundance of phyla play an essential role in stimulating the pathological process of Fritillaria ussuriensis.

We made the following analysis at the genus level of soil fungi and bacteria: for fungi, Mortierella was the most abundant genus in soil samples. Previous studies have shown that some species of Mortierella can produce antibiotics, and several isolates were used to develop antagonists to plant pathogens (Tagawa et al., 2010). In contrast, there have also been reports suggesting that certain species of Mortierella can cause disease; for example, Mortierella bainieri parasitizes Agaricus bisporus, which results in rough stipe (Palaniyandi et al., 2013). In this study, the relative abundance of Mortierella in diseased soils (P and B) significantly increased, therefore, we speculated that the relative abundance of pathogenic species in Mortierella genus in rhizosphere soil of Fritillaria ussuriensis was much higher than that of beneficial species. In future research, we will isolate these species in Mortierella and determine their role in the pathological mechanism of Fritillaria ussuriensis. The relative abundances of Fusarium and Humicola in diseased soils (P and B) significantly increased. Fusarium is a wilt-causing pathogen, and Humicola is the pathogenic fungus that causes root rot in plants (Ginetti et al., 2012). The relative abundance of these two genera is an important indicator for identifying Fritillaria wilt disease. Among the bacteria, the relative abundance of Sphingomonas increased significantly with increasing the severity of disease (H →P →B). Ali et al. (2019) found that Sphingomonas could promote plant growth by transforming organic matter, indicating that as the disease worsens, the relative abundance of Sphingomonas constantly increased, which resulted in antagonizing pathogen and maintaining the balance between rhizosphere soil microbes. The RB41 genus from the phylum Acidobacteria in the healthy soil had the highest relative abundance. Previous studies have shown that Acidobacteria could degrade lignin and cellulose to improve soil nutrients (Pankratov et al., 2011), indicating that the RB41 was dominant in the healthy soil of Fritillaria ussuriensis. The functions of Arthrobacter in efficiently degrading soil organic matter and alkaloids have been reported (Guo et al., 2019). The highest relative abundance of Arthrobacter in the healthy sample indicated that Arthrobacter contributes to soil nutrient cycling. It has been confirmed that Bacillus exerts a significant antagonistic effect on pathogens (Lin et al., 2020). We found a significant increase in the relative abundance of Bacillus after infection (P and B), indicating that this genus plays a critical role in the efficient inhibition of pathogenic microbes. The studies on disease-suppressive soils have shown that the plant root system would recruit beneficial microbes after being infected by pathogens, enhance biological activity, and inhibit pathogens (Raaijmakers et al., 2009). The pathogen-resistant soil is formed as a result of the long-term effect of soil infestation (Berendsen, Pieterse & Bakker, 2012), which indicates that strict regulations can be applied by accumulating some beneficial bacteria in diseased soils (P and B) of Fritillaria ussuriensis, which explains that the abundance of beneficial microbes, such as Sphingomonas and Bacillus, significantly increased with increasing the severity of disease (H →P →B). The highest relative abundances of RB41 and Arthrobacter in healthy soils indicated that these two bacterial genera had a great influence on maintaining the health of Fritillaria ussuriensis and ecological balance of rhizosphere soil.

Environmental factors have a significant impact on the diversity and structure of soil microbial communities (Zeng, Dong & An, 2016). Studies have shown that the plant abscissions and secretions could promote the diversity of the soil microbial community and the content of organic matter (Kuzyakov, Blagodatskaya & Blagodatsky, 2009). In this study, the Fritillaria ussuriensis have withered in large numbers at the end of the disease, so the content of organic matter in blank soil increased significantly, and microbial diversity also increased. On the other hand, soil salt content could directly inhibit the activity of microorganisms, meanwhile, influence the structure and composition of microbial community by changing soil fertility (Yan & Marschner, 2012). In our study, we found that the salt content of the diseased soils (P and B) increased significantly and the microbial diversity decreased significantly. There was a negative correlation between soil salt content and bacterial diversity, which was consistent with the research results of Liu et al. (2021) on cotton soil bacterial communities at different disease levels. Changes in soil nutrients could affect the composition of microbial communities (Feng et al., 2019), and enrichment of nutrient elements could increase the number of pathogenic microorganisms, leading to an increase in plant disease rates (Song et al., 2017). For example, Liu et al. (2020) showed that when the N content in the soil was low, the incidence of wheat root rot was significantly decreased, and it was closely related to the decrease of Fusarium. We also found that TN, AN, TP, AP, TK and AK were significantly increased in disease (P and B) soil through data analysis, and they were positively correlated with the abundance of pathogenic microorganisms such as Fusarium and Humicola et al.. In addition, soil pH was an important factor that determined the diversity and structure of soil microbial communities (Bainard, Hamel & Gan, 2016; Meng et al., 2019). Studies have shown that fungi were more common in acidic soils (Rousk, Brookes & Baath, 2009), but when the pH value increased, the soil microbial diversity also increased, the bacterial growth rate increased, and the bacterial community composition changed (Baath & Arnebrant, 1994), which explained the positive correlation between soil pH and bacterial diversity in this research. It can be seen that changes in environmental factors have caused changes in the number, diversity and structure of soil microorganisms, and the correlation between environmental factors and soil microorganisms affects the health of Fritillaria ussuriensis.

Microbial ecological networks revealed distinct patterns of the microbial community in the rhizosphere soil of Fritillaria ussuriensis. In the fungal networks, Fusarium and Humicola were dominant, yet most sequences of Fusarium were not affected by the wilt pathogenic species (Fusarium oxysporum), which can induce disease (Wu et al., 2017). High incidence of disease may be associated with an increased abundance of other congeneric species, indicating that the microbial interactions have more influence on plant health. The correlation-based bacterial network was more complicated than that of fungi. Previous studies have shown that more interactions can promote cooperation in the complex microbial community (Zhang et al., 2014). Although high levels of cooperation might be linked to a higher function of the community, such interactions can also cause destabilization (Coyte, Schluter & Foster, 2015). Highly connected networks can stabilize the soil microbial community and improve the overall resistance to pathogens (Scheffer et al., 2012). Bahram et al. (2018) have shown that both the environmental factors and microbial correlation could affect the diversity and structure of microbial communities. Eldridge et al. (2015) have shown that the difference in the correlation between fungi and bacteria in the soil microbial ecological network was mainly determined by the degree of soil interference, and environmental factors could increase the instability of the microbial community structure. Therefore, changes in environmental factors affected the soil microbial correlation, environmental factors and microbial correlation complement each other, and together affected the health of plants. Based on the impact of environmental factors on microorganisms and the microbial association network, correlation-based variations of microbes in the rhizosphere soil at different health levels can be understood. This can lay a foundation for the systematic study on the interaction between microbial genera in the pathological process of Fritillaria ussuriensis.

This research systematically explained the physicochemical properties, the diversity and structure of microbial communities, the correlation between environmental factors and soil microorganisms, and the co-correlation network among microorganisms in the rhizosphere soil of Fritillaria ussuriensis at different health levels. However, there were some limitations in our study. First of all, as an endemic species in Northeast China, Fritillaria ussuriensis had certain limitations in the planting range and growth environment. In addition, this study did not involve research on the effects of Fritillaria ussuriensis root exudates in different health levels on soil physicochemical properties and soil microbial communities. Through high-throughput sequencing, the dominant populations of soil microorganisms in different health levels were determined, but they were not isolated, and pathological studies were not systematically carried out. In future research, we will make up for these shortcomings, isolate and cultivate these beneficial microorganisms in Fritillaria ussuriensis rhizosphere soil for disease resistance research. The above work will be of great significance to the prevention of Fritillaria ussuriensis wilt disease and the maintenance of a healthy soil micro-ecosystem.

Conclusion

The physicochemical properties and microbial community diversity of Fritillaria ussuriensis rhizosphere soils in different health levels were significantly different. Compared with the healthy soil, the diversity of diseased soils (P and B) showed a decreasing trend. There were also significant differences in the composition of microbial communities in rhizosphere soils of Fritillaria ussuriensis at different health levels. The relative abundance of Fusarium and Humicola in diseased soils (P and B) was significantly increased, while in healthy soils, the relative abundance of RB41 and Arthrobacter was the highest. These soil microorganisms affect the health level of Fritillaria ussuriensis through a close and complex relationship network. At the same time, this research revealed that the differences in microbial communities from the rhizosphere soils of Fritillaria ussuriensis were the key factors that caused changes in environmental factors. In future research, we will isolate beneficial microorganisms from healthy rhizosphere soil samples and research the mechanism of the beneficial microorganisms growth promotion and disease resistance in Fritillaria ussuriensis. In addition, dominant fungi Mortierella in present research needs further research to determine the role of key species in the pathological mechanism of Fritillaria ussuriensis. These work will provide an important basis for the rhizosphere soil micro-ecology restoration and wilt disease prevention of Fritillaria ussuriensis.

Supplemental Information

Supplemental Information 1 Comparative analysis of soil physicochemical properties of Healthy, Pathology and Blank

Different letters indicate a significant difference at P < 0.05 according to Duncan’s new multiple range test. OM, organic matter; TN, total nitrogen; AN, available nitrogen; TP, total phosphorus; AP, available phosphorus; TK, total potassium; AK, available potassium; salt, total soluble salt.

Click here for additional data file.

Supplemental Information 2 Correlation analyses between diversity indices and soil properties

∗ and∗∗ represent significance (P < 0.05 and P < 0.01) of soil samples according to Pearson’s correlation analysis.

Click here for additional data file.

Supplemental Information 3 Rarefaction curves of Healthy, Pathology and Blank samples (97% sequence similarity)

Click here for additional data file.

Additional Information and Declarations

Competing Interests

Author Contributions

Data Availability

The authors declare there are no competing interests.

Ning Jiao and Xun Deng conceived and designed the experiments, performed the experiments, analyzed the data, prepared figures and/or tables, authored or reviewed drafts of the paper, and approved the final draft.

Xiaoshuang Song, Ruiqing Song and Dachuan Yin analyzed the data, authored or reviewed drafts of the paper, and approved the final draft.

The following information was supplied regarding data availability:

The sequence data are available in the NCBI Sequence Read Archive: SRR13288220 –SRR13288237.

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
