# Peer review of "Diversity and structure of the microbial community in rhizosphere soil of Fritillaria ussuriensis at different health levels"

_PeerJ, doi:10.7717/peerj.12778_

## Round 0.1 · original submission · Major Revisions

Dear Dr. Jiao and co-authors,

I just received the reviews of your manuscript. Please, consider all comments and suggestions provided by both reviewers during the revision of your manuscript.

A comprehensive revision of the English of the manuscript is necessary before submitting the new version.

Don't forget to include a letter response along with the revised version of the manuscript. In this letter you must respond point by point to each question.

Best regards,

Xiaoming Kang

Reviewer 1 ·

Basic reporting

no comment

Experimental design

1 The health status of Fritillaria ussuriensis varied greatly in different plots. Please describe the geographical differences between different points.

2 How to distinguish between three different levels of health of rhizosphere soils. Please explain this in manuscript.

3 In figure 7, the significantly higher abundances of Massilia and Fusarium with stronger correlations with other genera were found, but couldn't indicat that these two genera played a dominant role in the pathogenesis of Fritillaria ussuriensis. Which is more important, environmental factors or microbial factors?

4 Please add references about the explain that "The soil environment could have been changed due to the death and decay of a large number of Fritillaria ussuriensis plants, which in turn affected the variation in the soil microbial community."

5 Is it the co-correlation networks or co-existence networks?

Validity of the findings

no comment

Additional comments

no comment

Reviewer 2 ·

Basic reporting

Introduction section:
Please revise the section of introduction to clarify these issues:
1) There is little here which is intrinsically novel or insightful.
2) Why should this research be conducted on the study site (i.e., Hailin Forestry Farm)?
3) There are no robust hypotheses or mechanistic insights provided which further inform the discipline of soil biochemical cycles in a general sense. In short, the authors are not unravelling relevant mechanisms that reported from other researches, nor put forward their own hypotheses.
5) Line 51-54: Please add the references here.

Experimental design

Materials and methods section:
1) A map with geographical location of the study site will be more intuitive for readers.
2) Please provide some basic information on soil (e.g., the soil water content, temperature and soil porosity), the specific sampling date or season and the course of the disease.
3) Please describe clearly the differences among the health, pathology and blank levels. It would be better to provide some pictures that present different levels.

Validity of the findings

Results section:
1) Line 108: The rarefaction curve was just used to identify the validity of the data, and it was a stage of data processing, I recommend to put the Figure 1 into the supplement materials.
2) Line 217-220: Which sample did the data refer to? Please rephrase these sentences to clarity your meaning.
3) Figure 5: ‘bacterial phyla’ change to ‘bacterial genera’. Please mark the differences among the three disease levels and unify data format of P-value.
4) Is there any difference in the fungal and bacterial community network between the diseased sample and the healthy sample?
5) Line 295-298: Please rephrase this sentence to clarity your meaning.
6) I recommend to use the structural equation modelling (SEM) to evaluate the relationships among soil properties, microbial communities and treatments.

Discussion section:
The logic of this section was poor, please revise as suggested below:
1) Line 300-309: Delete these sentences as they are the content of introduction part.
2) Line 320-323: This sentence is confused, please rephrase it to clarity your meaning.
3) Line 324-326: The ‘soil microbial community’ was not appropriately used here, please change it to microbial diversity or a specific index.
4) Line 330-333: The author report: ‘…. the soil microbial diversity was related to the differences in crop species, soil types, and environmental factors…’, How are they related with each other, and why, please clarify them in detail or set some examples.
5) Line 343-346: Same issue with 4), please clarify your point in detail.
6) Line 365-366: Delete this sentence, it is useless.
7) Line 371: Should be ‘relative abundance’, and be careful to check the use of ‘abundance’ and ‘relative abundance’ in the whole MS, and revise the mistake.
8) Line 371-375: What is the authors’ intention of this sentence?
9) Line 384-386: This sentence is confused, please rephrase it to clarity your meaning.
10) Line 392-394: Please add the references here.
11) Line 402-433: This paragraph should illustrate the mechanisms with respect to how environmental factors affect the microbial community structure in different levels of disease. Obviously, it failed. The authors only provided a series of related researches (some of them may be not appropriate) and make the comparison with them. Please revise this paragraph carefully.
12) In the Discussion section, the author should also reply the important problems: what are the limitations of this study and what are the implications for future research, such as the prevention of the disease.

Conclusion section:
The conclusions are not at all conclusive, but yet another summary of the findings and other content in the paper. Please revise this section.

Additional comments

no comment

Reviewer 3 ·

Basic reporting

English writing is terrible and the authors should substantially improve. the background of the research is provided. the article structure is reasonable.

Experimental design

the research question is meaningful for the understanding of the effects of soil microbial community on plant health. The experimental method is suitable for this study and could provide support for the conclusion. the authors should explain why did not determine the bacterial and fungal diversity using the same soil sample at each healthy level.

Validity of the findings

the main findings of this study should be refined. the authors should tell what is the most important finding in the conclusion section.

Additional comments

I suggest that the authors should refine the main finding of this research and tell a attractive story. In addition, the authors should substantially improve English writing and I therefore recommend it for publication after major revision.

Annotated reviews are not available for download in order to protect the identity of reviewers who chose to remain anonymous.

---

## Round 0.2 · accepted · Accept

Dear authors,

I am pleased to inform you that, following the revision made based on the reviewer's comments, your manuscript is now acceptable for publication in PeerJ.

Best regards

Xiaoming Kang

Reviewer 2 ·

Basic reporting

no comment

Experimental design

no comment

Validity of the findings

no comment

Additional comments

This manuscript has greatly improved following the careful responses to the first round of reviews. I don't have any further concerns.

Reviewer 3 ·

Basic reporting

The English writing has been improved.

Experimental design

Research method section has also been revised.

Validity of the findings

The conclusions have been refined.